# Separating Landslide Source and Runout Signatures with Topographic Attributes and Data Mining to Increase the Quality of Landslide Inventory

## Jhe-Syuan Lai

Department of Civil Engineering, Feng Chia University, Taichung 40724, Taiwan; jslai@fcu.edu.tw;
Tel.: +886-4-2451725 (ext. 3118)

**Abstract:** Landslide sources and runout features of typical natural terrain landslides can be observed from a geotechnical perspective. Landslide sources are the major area of occurrences, whereas runout signatures reveal the subsequent phenomena caused by unstable gravity. Remotely sensed landslide detection generally includes runout areas, unless these results have been excluded manually through detailed comparison with stereo aerial photos and other auxiliary data. Areas detected using remotely sensed landslide detection can be referred to as "landslide-affected" areas. The runout areas should be separated from landslide-affected areas when upgrading landslide detections into a landslide inventory to avoid unreliable results caused by impure samples. A supervised data mining procedure was developed to separate landslide sources and runout areas based on four topographic attributes derived from a 10–m digital elevation model with a random forest algorithm and cost-sensitive analysis. This approach was compared with commonly used methods, namely support vector machine (SVM) and logistic regression (LR). The Typhoon Morakot event in the Laonong River watershed, southern Taiwan, was modeled. The developed models constructed using the limited training data sets could separate landslide source and runout signatures verified using the polygon and area constraint-based datasets. Furthermore, the performance of developed models outperformed SVM and LR algorithms, achieving over 80% overall accuracy, area under the curve of the receiver operating characteristic, user's accuracy, and producer's accuracy in most cases. The agreement of quantitative evaluations between the area sizes of inventory polygons for training and the predicted targets was also observed when applying the supervised modeling strategy.

**Keywords:** data mining; landslide detection; landslide inventory; Typhoon Morakot

## 1. Introduction

Natural hazards occur frequently in Taiwan. Among them, landslides can be triggered by heavy rainfall events, especially in mountainous areas. The precipitous terrain, complicated geology, and dense population may increase vulnerability, causing a serious loss of life, property damage, and economic loss. Furthermore, these events can affect water resource supply and cause other livelihood problems. Therefore, landslide analysis and assessment have become critical in natural hazard and disaster mitigation, prevention, and reconstruction in Taiwan. A growing number of studies have investigated landslide-related topics on different scales, such as slope stability analysis for a specific slope site [1,2]; regional landslide detection and mapping [3–7]; characterization of the relationship between landslides and environmental or triggering factors [8–13]; susceptibility, hazard, risk assessment, and management [14–19]; and modeling or estimating physical, environmental, and rainfall-based parameters [20–24]. Landslide risk assessment and management are crucial, systematic, and extensive frameworks in the related works. Dai et al. [14] divided this topic into

several parts. In this framework, generating landslide inventory (inventory map and database are used synonymously in this study) is essential to connect the following assignments.

A basic landslide inventory should record the landslide's location, date (event-based) or period (multitemporal), and types of movement [25]. The definition, assumptions, requirements, production processes, and statistical properties of landslide inventories have been discussed in detail by Guzzetti et al. [25], Harp et al. [26], Malamud et al. [27], and Shao et al. [28]. Furthermore, Galli et al. [29] and Mondini et al. [11] have assessed the quality and completeness of landslide inventory maps by comparing two maps of the same area. Numerous studies have indicated that the continuing improvements in remote sensing and geographic information systems (GISs) have led to cooperation with data mining and machine learning algorithms to produce a regional landslide inventory. In particular, integrating GIS-based models with geo-spatial data [18,30] and generating event-based landslide inventory [11,25,31] have garnered interest.

Three common features of typical natural terrain landslides have been observed from a geotechnical perspective. These features were the landside source, trail, and deposition fan [32]. The surface of the rupture, comprising the main scarp and the scarp floor, is defined as the source area. A landslide trail may also occur predominantly as a result of the transport of the landslide mass. The majority of the landslide mass is deposited (i.e., deposition fan). The term runout is generally used to indicate the landslide trail and deposition fan [33]. Landslide detection performed using automatic and semiautomatic methods with remotely sensed images usually includes runout areas unless these results have been excluded manually through a detailed comparison with stereo aerial photos and other auxiliary data. The areas detected using remotely sensed landslide detection can be referred to as "landslide affected" areas. Mixing landslide source areas with runout regions may reduce the reliability of a landslide analysis [34], such as susceptibility and hazard assessments. These runout areas should be separate from landslide source areas because they have different mechanisms. More precisely, producing landslide inventory requires further processing after the remote sensing of landslide detections.

Data mining and machine learning-based algorithms have been used increasingly for landslide modeling, especially in landslide susceptibility assessments, such as decision trees, deep learning systems, evolution-based algorithms, fuzzy theory, neural networks, random forests (RF), and support vector machines (SVM) [35–40]. Related studies have also considered various composites and compared the effectiveness of strategies based on established methods in achieving a specific purpose [41–43]. However, few studies have employed these approaches for detecting landslides and producing inventories that consider the separation of runout areas from landslide affected regions. The RF method [44] consists of data mining and machine learning algorithms that have displayed excellent performance in the analysis of numerous remote sensing and landslide topics [45,46]. The RF algorithm is based on the ensembles of various decision tree results and exhibit desirable properties, such as high accuracy, robustness against overfitting the training data, and integrated measures of variable importance [47]. Furthermore, Lai and Tsai [48] and Lai et al. [34] have demonstrated that combining the RF algorithm with a cost-sensitive analysis [49] in landslide susceptibility assessments can reduce extreme omission (missing) or commission (false alarm) predictions because it adjusts the decision boundary.

The main purpose of this study was to explore the feasibility of separating landslide source and runout areas based on landslide affected extents extracted from remotely sensed images in order to upgrade landslide detections into a landslide inventory. The significance of topographic data in relation to the flow velocity, geomorphology, runoff rate, soil water content, and differences between landslide source and runout areas was demonstrated [34,50]. The term "signature" used in this study represents the patterns of landslide source and runout areas in a feature space. Feature space means that the dimension is the used factors with the samples based on a scatter plot form. Therefore, the developed RF-based data mining models and topographic attributes, such as aspect, curvature, elevation, and slope, constructed for the Typhoon Morakot event in 2009 were combined to compare the results with those

obtained using SVM and logistic regression (LR) algorithms. SVM and LR algorithms are commonly applied in the related domains. The performance of cost-sensitive analysis was also evaluated to improve the constructed models by adjusting the decision boundary.

## 2. Materials and Methods

### 2.1. Study Site and Data

The study area was located in the Laonong River watershed in southern Taiwan and covered approximately 117 km$^2$, as illustrated in Figure 1a. The elevation in the study site ranged from 258 to 1666 m above sea level, measured using the digital elevation model (DEM) modified by Chiang et al. [51]. The average elevation was 716 m. The slope range, average slope, and standard deviation were 0–71.2°, 25.84°, and 11.98°, respectively, which indicated that the terrain of the Laonong River watershed is steep. Three geological formations and four soil types were identified in the study area. The geological formations were Lushan, Sanhsia, and Toukoshan. The four soil types were alluvium, colluviums, lithosol, and loam. Lai et al. [34] reported detailed information regarding geological formations and soil types within the study site. The Laonong River watershed is located in the tropical monsoon region, and the annual precipitation is approximately 3400 mm. Therefore, this area is frequently struck by typhoons. For example, Typhoon Morakot caused extensive rainfall in 2009, which resulted in numerous landslides and debris flow in southern Taiwan. An official report [52] indicated that 769 casualties or missing people were directly or indirectly caused by these landslides. Furthermore, Typhoon Morakot caused losses of approximately $526 million because of the damage to agriculture, forestry, and fishery. In particular, a riverside village called Xiaolin (sometimes spelled Shiaolin, Hsiaolin) was destroyed by the landslides and debris flows from a devastating landslide nearby, which caused approximately 500 fatalities.

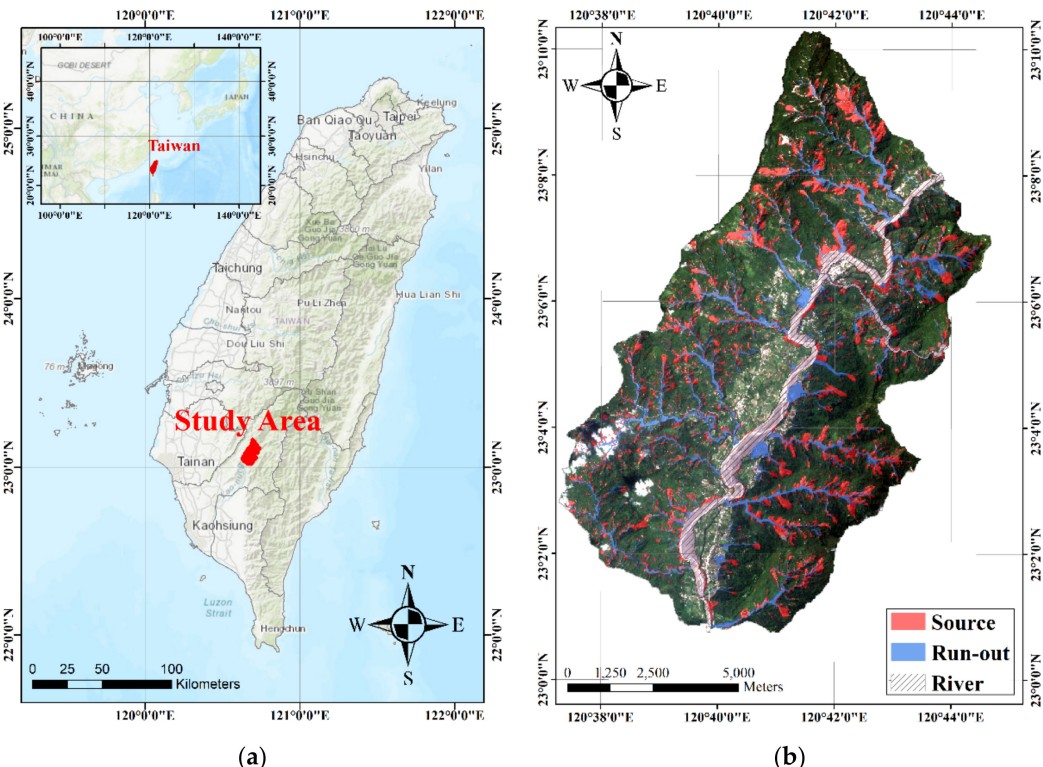

(a)　　　　　　　　　　　　　　　　　　　　(b)

**Figure 1.** Location of the study site (**a**) and landslide inventory map (**b**).

A landslide inventory map of Typhoon Morakot revealed deep-seated landslides, as illustrated in Figure 1b. The landslide source, runout, and channel classes were interpreted manually [34] based

on stereo aerial photos and auxiliary data. This study assumed that the used landslide inventory comprised the results of the detection of affected landslide areas extracted from the remotely sensed images. The landslide inventory was also used for quantitative verifications. The 10–m DEM was then analyzed to derive topographic attributes, including the aspect, curvature, elevation, and slope, as illustrated in Figure 2. Furthermore, the landslide inventory map was converted into a grid format of 10 m² to match the topographic factors for constructing data mining-based models.

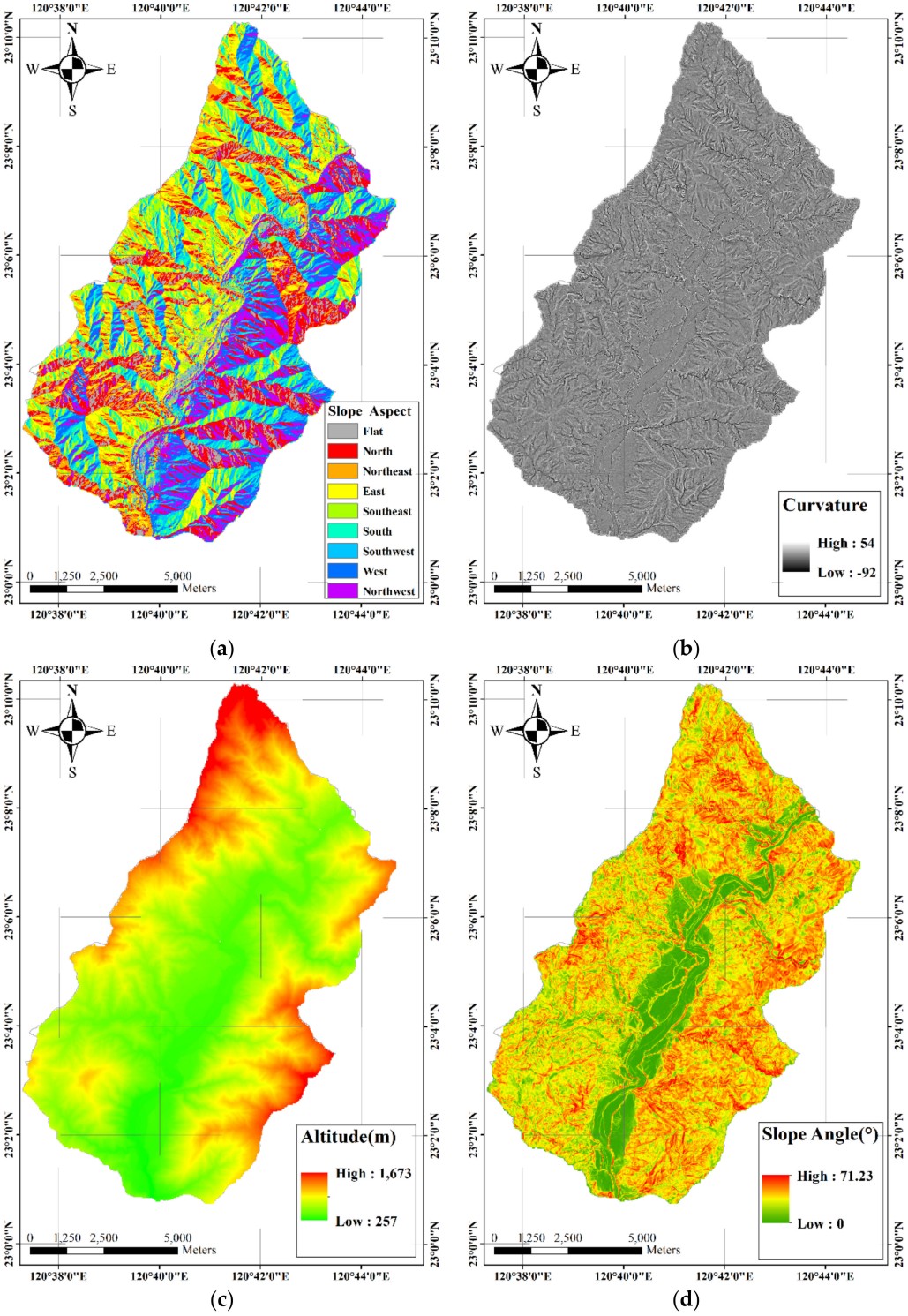

**Figure 2.** Topographic layers used in this study. (**a**) Aspect, (**b**) curvature, (**c**) elevation, and (**d**) slope.

## 2.2. Procedure and Data Mining Algortithms

Figure 3 illustrates the conceptual procedure implemented in this study. The topographic attributes were derived from the 10–m DEM, which connected the samples extracted from the landslide inventory for modeling (Sections 2.2.1 and 2.2.2) and verification (Section 2.3). Two experiments were designed for the demonstration of separating landslide source and runout signatures. First, the limited training datasets were selected (the five largest area sizes of landslide source and runout classes in the landslide inventory polygons) for polygon by polygon modeling. Second, each model was verified using other four polygon-based samples. Third, these models were used to predict the polygon samples with an area of ≥100,000 and 50,000–100,000 m$^2$ to identify the optimal model. Previous results are presented in Sections 3.2 and 3.3. Detailed information regarding the samples extracted from the landslide inventory is presented in Table 1. Finally, the spatial patterns of landslide source and runout areas were produced instance by instance based on the output labels of the optimal model, as demonstrated in Section 3.4.

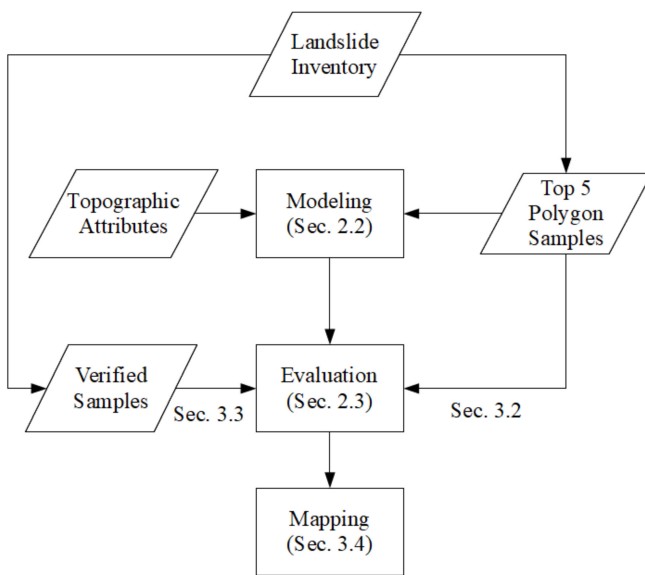

**Figure 3.** The conceptual procedure used in this study to model and verify the separation between landslide source and runout signatures.

**Table 1.** The training and verification samples extracted from the landslide inventory polygons.

| Dataset | Landslide Source | | Runout | |
|---|---|---|---|---|
| | Area (m$^2$) | No. of Samples | Area (m$^2$) | No. of Samples |
| No. 1 | 250,493 | 2508 | 385,381 | 3839 |
| No. 2 | 154,709 | 1545 | 239,199 | 2386 |
| No. 3 | 141,574 | 1412 | 206,055 | 2060 |
| No. 4 | 132,005 | 1313 | 197,162 | 1980 |
| No. 5 | 126,636 | 1265 | 190,338 | 1905 |
| ≥10,000 m$^2$ * | 155,333 (avg) | 9318 | 162,768 (avg) | 27,640 |
| 50,000—100,000 m$^2$ | 68,670 (avg) | 15,798 | 63,792 (avg) | 8937 |

*Areas and number of samples in the table include the training polygon samples, but these samples were eliminated during the verified process.

### 2.2.1. Developed Models

The random forests (RF) algorithm [44] is a data mining method used for constructing a classification-based model that employs a supervised strategy. RF is an extension of the tree-based algorithms. The RF employs the information gain (IG) measure or Gini index to determine the degree of impurity of factors or variables, applying the bootstrap and vote operators to improve the results

derived from the tree-based rules. The bootstrap method "randomly" selects subtraining data to generate several trees (termed "forests") to avoid the overfitting results. The vote component is used to determine the optimal results according to the developed trees, thus improving the RF classifier. A topographic factor with a larger IG or Gini value had to be selected because higher values indicate a higher priority for constructing a tree node, which should be ignored in the next computation. A sequence of tree-based rules was constructed after several iterations. The rules were then used to classify other instances. Nominal (or discrete) and numeric (or continuous) formats are commonly used in the field. Entropy theory was applied to calculate the IG in the nominal case, as displayed in Equations (1)–(3). In these equations, E(A) represents the entropy of all training data; S and R represent the numbers of landslide sources and runout samples, respectively; E'(a) and v are the entropy and number of subsets of a specific topographic factor, respectively; E(aj) is the entropy of the subset in a specific topographic factor, calculated using Equation (1); IG(a) represents the IG of a specific topographic factor. The Gini index was applied to determine the IG in the numeric cases, as displayed in Equations (4) and (5), where C represents a segmented point for a specific topographic factor used to divide continuous data into two parts, and $N_1$ and $N_2$ represent the numbers of a $\leq$ C and a > C, respectively. The steps were detailed by Guo et al. [53].

$$E(A) = -\frac{S}{S+R}\log_2\frac{S}{S+R} - \frac{R}{S+R}\log_2\frac{R}{S+R} \tag{1}$$

$$E'(a) = \sum_{i=1}^{v} \frac{S_i + R_i}{S+R}E(a_i) \tag{2}$$

$$IG(a) = E(A) - E'(a) \tag{3}$$

$$Gini(a \leq C \text{ or } a > C) = 1 - \sum_{i=1}^{m}\frac{n_i}{N} \tag{4}$$

$$IG(a,C) = \frac{N_1}{N}Gini(a \leq C) + \frac{N_2}{N}Gini(a > C) \tag{5}$$

The strategy of adjusting the decision boundary used in the study is termed as the cost-sensitive analysis. The cost-sensitive analysis is a postclassification method based on the cost matrix that reclassifies the instances and balances the accuracies of certain classes when missing or false alarm errors are unreliable [49,54,55]. The dimension of the cost matrix is equal to that of the confusion matrix. The confusion matrix used in this study is presented in Table 2. True negative (TN), false negative (FN), false positive (FP), and true positive (TP) were used to represent an agreement between the classified and reference labels in counts tabulated in a confusion matrix. The cost in this study indicated a weighting without the unit. The diagonal costs in the table represent correct results for the TN and TP, and the remaining costs indicate misclassification costs between the FN and FP. The diagonal costs and other elements are usually set to 0 and 1, indicating unadjusted and adjusted conditions of the decision boundary. A large amount of missing or false alarm errors are severe FP or FN errors, respectively. Adjusting the decision boundary by increasing the cost (weighting) of the FP or FN leads to the inclusion of more samples, thereby balancing the classification result of a certain class. Equation (6) displays the optimal prediction (R) of sample x in class i, where P(j|x) is the likelihood of estimating a classification of a sample into all classes (j), and C represents the costs. In binary cases, the optimal prediction is the landslide source label (class 1 or positive) if the expected cost of this prediction is less than or equal to the expected cost of predicting the runout label (class 0 or negative), as displayed in Equation (7). Given p = P(1|x) and $C_{TN} = C_{TP} = 0$, Equation (7) can be simplified, as displayed in Equation (8). An adjusted threshold (p*) of the decision boundary, presented in Equation (10), can be

derived from Equation (9) to classify a sample x as the landslide source label when P(1|x) is larger than or equal to the threshold [56]:

$$R(i|x) = \sum_j P(j|x)C_{ji} \tag{6}$$

$$P(0|x)C_{FP} + P(1|x)C_{TP} \leq P(0|x)C_{TN} + P(1|x)C_{FN} \tag{7}$$

$$(1-p)C_{FP} \leq p\,C_{FN} \tag{8}$$

$$(1-p^*)C_{FP} \leq p^*C_{FN} \tag{9}$$

$$p^* = \frac{C_{FP}}{C_{FP} + C_{FN}} \tag{10}$$

**Table 2.** The confusion matrix used in this study.

| Confusion Matrix | | Ground Truth | |
|---|---|---|---|
| | | **Runout** | **Landslide Source** |
| Prediction | Run-out | True Negative (TN) | False Negative (FN) |
| | Landslide source | False Positive (FP) | True Positive (TP) |

### 2.2.2. Algorithms for Comparison

Two commonly used data mining methods were used for comparison with the developed models. The support vector machines (SVM) method has been wieldy applied in remote sensing studies [57] to classify land cover or use targets, extract biophysical parameters in different spatial resolutions, and perform landslide analyses [40,41]. The core role of the SVM classifier is to determine the largest margin of the classifier based on factor transformations with linear and nonlinear functions. These transformations are known as the kernel function. Equation (11) describes a concept of the SVM-based classification with a "sign" operator for the binary classification, where l, y, and k(xi,xj) indicate the number of used factors (support vectors), the outputted label, and a kernel function, respectively, and a and b represent the coefficients of the margin. The kernel functions used in this study were linear, polynomial, and radial basis functions.

$$f(x) = \text{sign}\left[\sum_{i=1}^{l} a_i y_i k\left(x_i,\ x_j\right) + b\right] \tag{11}$$

Logistic regression (LR) is a typical and traditional statistical method used to model landslide events [20,58]. The computation describes the relationship between the dependent and independent variables as defined in Equation (12), where p refers to the likelihood or probability, a is a constant, b is the regression coefficient, and x represents independent variables or used factors:

$$p = \frac{\exp(a + b_1 x_1 + b_2 x_2 + \ldots + b_n x_n)}{1 + \exp(a + b_1 x_1 + b_2 x_2 + \ldots + b_n x_n)} \tag{12}$$

### 2.3. Accuracy Assessment

The quantitative results of comparing the output and ground truth labels, identified based on the threshold, were derived from the confusion matrix to verify the constructed models. An instance was categorized into the landslide (positive) class in most cases when the likelihood of occurrence was ≥0.5; otherwise, the instance was assigned the nonoccurrence (negative or non-landslide) label. Similar to previous classifications, the positive and negative labels in this study were changed from landslide and non-landslide to landslide source and runout classes, respectively. The threshold of 0.5 for classification could also be adjusted by the cost-sensitive analysis (p*).

Four commonly used quantitative indices derived from a confusion matrix, namely overall accuracy (OA), user's accuracy (UA), producer's accuracy (PA), and area under the receiver operating characteristic (ROC) curve (AUC), were used to quantitatively evaluate the developed models. OA represents the ratio of samples correctly classified as TN and TP, as illustrated in Equation (13). UA and PA reflect the errors for each class (landslide source and runout) as presented in Equations (14)–(17), where R and S represent the runout and landslide source classes. The false alarm (commission errors) and missing (omission errors) assignments can be calculated as 1–UA and 1–PA, respectively. The AUC is commonly used to assess data mining-based models. The binary classification threshold is used for reclassifying instances, calculating the rates of FPs and TPs to draw the ROC curve. In general, the AUC ranges from 0.5–1, and results larger than 0.7 are acceptable, and larger than 0.8 are excellent [59].

$$OA = \frac{TN + TP}{TN + FN + FP + TP} \tag{13}$$

$$UA(R) = \frac{TN}{TN + FN} \tag{14}$$

$$UA(S) = \frac{TP}{FP + TP} \tag{15}$$

$$PA(R) = \frac{TN}{TN + FP}. \tag{16}$$

$$PA(S) = \frac{TP}{FN + TP} \tag{17}$$

## 3. Results

The data mining modeling and accuracy assessments used in this study were developed using the WEKA environment (http://www.cs.waikato.ac.nz/mL/weka/), which is a free and open-source platform. A four-stage step was designed to explore the separability between landslide sources and runout signatures. The first stage, as reported in Section 3.1, compared the topographic attributes between the landslide source and runout samples extracted from different sizes of landslide inventory polygons that had an area of ≥100,000 or 50,000–100,000 m$^2$. In the second stage, the RF and cost-sensitive analysis-based data mining algorithms were used to construct the models polygon by polygon, with the limited training datasets (the five largest area sizes in the landslide inventory polygons were considered), as reported in Section 3.2. The major parameter in the RF-based computation was the number of trees (Ntree). Du et al. [60] determined that 10–200 Ntree did not have any effect on the results, but an increase in Ntree increased the computation loading. Therefore, 100 trees were adopted in this study, as suggested by WEKA, to develop the RF-based models. In the third stage (Section 3.3), the constructed models were applied to predict other landslide source and runout samples extracted from the polygons with an area of ≥100,000 and 50,000–100,000 m$^2$, respectively. The developed models were compared with the commonly used methods, namely SVM and LR. The fourth stage consisted of visualizing the spatial distributions of predicting the landslide source and runout areas to assess the performance of the constructed model, as reported in Section 3.4.

### 3.1. Signatures of Topographic Attributes

The constraints on the area size of the inventory polygons were compared in the pairs of the topographic attributes to preliminarily examine differences between landslide source and runout signatures. The samples were extracted based on area sizes of ≥100,000 and 50,000–100,000 m$^2$. Furthermore, the value domain of the aspect, curvature, elevation, and slope layers was normalized into the range of 0–1 for the visualization of data distribution and separation. The patterns obtained by comparing the topographic datasets and area constraints are illustrated in Figures 4 and 5, respectively.

A larger area size enabled a larger separation of landslide source and runout features. The disagreement was not observed in the smaller landslide inventory polygons. This trend is similar to findings reported by Lai et al. [34]. This trend may reflect that the complete mechanisms of landslide source and runout areas appear in the larger landslide sizes. Moreover, the elevation is a critical factor in this study. These data distributions reveal the feasibility of distinguishing the landslide source and runout signatures when the sizes of landslide-affected areas were sufficient to reveal their mechanisms.

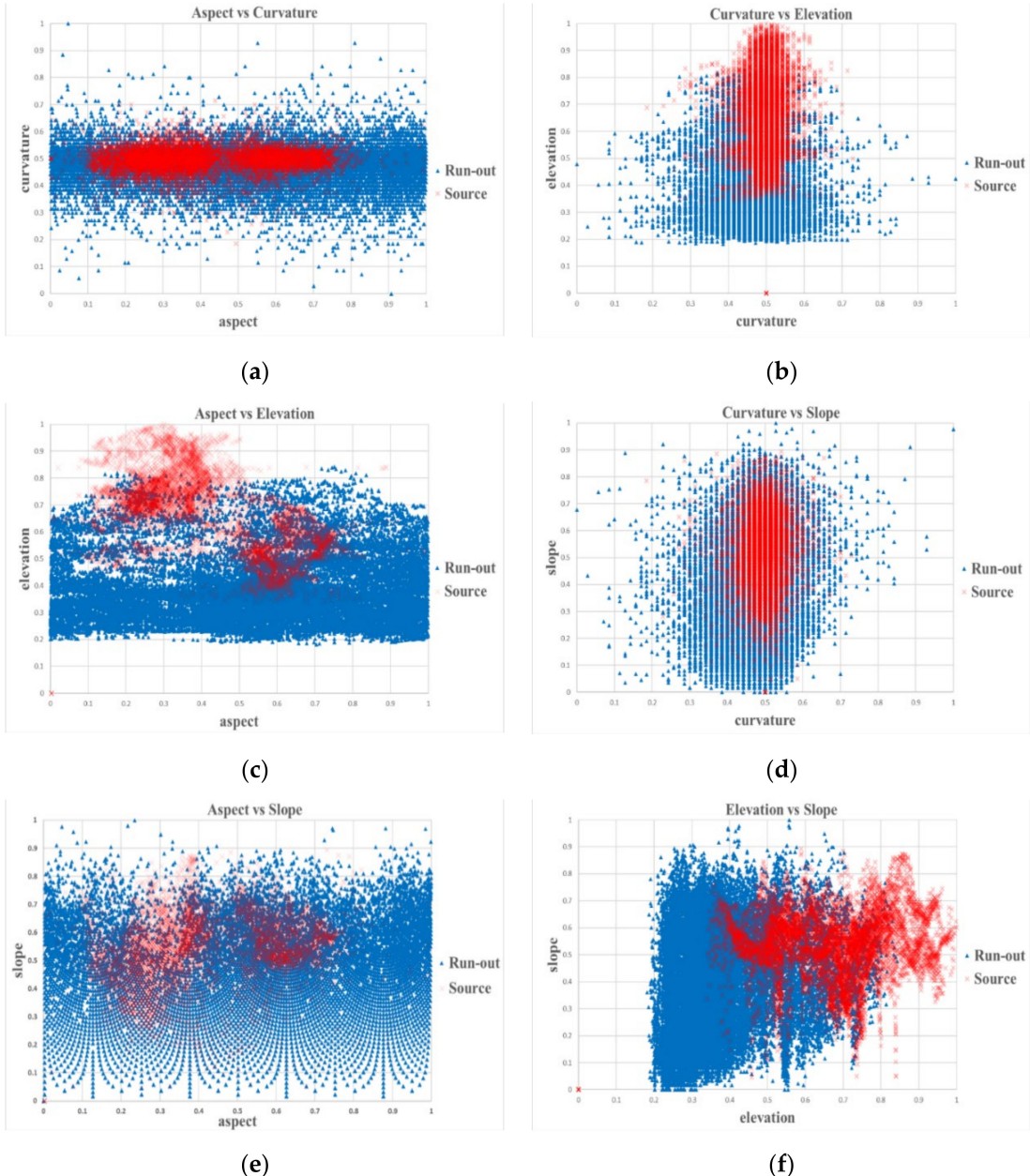

**Figure 4.** Patterns for comparing aspect, curvature, elevation, and slope layers with the samples extracted from the inventory polygon with an area of ≥100,000 m$^2$, where red and blue dots represent the landslide source and runout classes, respectively. (**a**): Aspect vs. curvature.; (**b**): curvature vs. elevation; (**c**): aspect vs. elevation; (**d**): curvature vs. slope; (**e**): aspect vs. slope; (**f**): elevation vs. slope.

*3.2. Construction of Random Forests Based Data Mining Models*

The RF-based data mining models were constructed using the top five polygons of area sizes in the landslide inventory polygons to further explore the feasibility of separating the landslide source and

runout areas. Each polygon was transformed into a grid format of 10 m$^2$. These samples could be used to extract the corresponding topographic attributes and produce the training and verification datasets. The quantitative evaluations of OA, AUC, PA, and UA are presented in Figure 6. The accuracies exceeded 80% in most cases. Model No. 3 provided the most accurate predictions. However, the results of predicting (a) the No. 3 and No. 5 polygon samples by using the No. 1 and No. 2 models, (b) the No. 5 polygon samples by using the No. 3 and No. 4 models, and (c) the No. 1 and No. 2 polygon samples by using the No. 5 model revealed a disagreement between the training and verification datasets. Possible reasons for this disagreement include (a) imbalanced sample ratios between landslide source and runout classes, (b) dissimilar data distributions between different inventory polygons, (c) lack of other critical factors, and (d) limited training data.

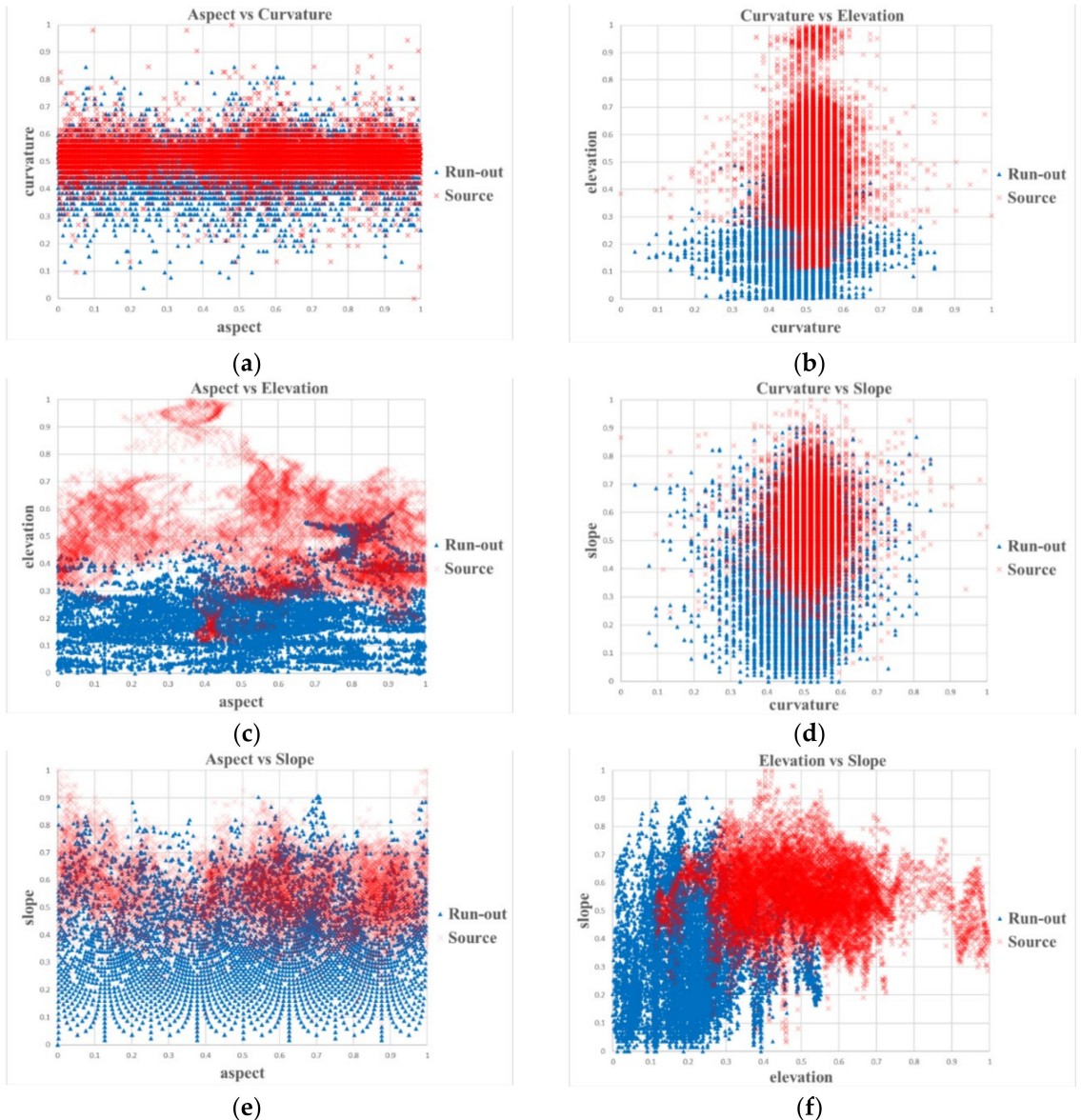

**Figure 5.** Patterns for comparing aspect, curvature, elevation, and slope layers with the samples extracted from the inventory polygon with an area of 50,000–100,000 m$^2$, where red and blue dots represent the landslide source and runout classes, respectively. (**a**): Aspect vs. curvature.; (**b**): curvature vs. elevation; (**c**): aspect vs. elevation; (**d**): curvature vs. slope; (**e**): aspect vs. slope; (**f**): elevation vs. slope.

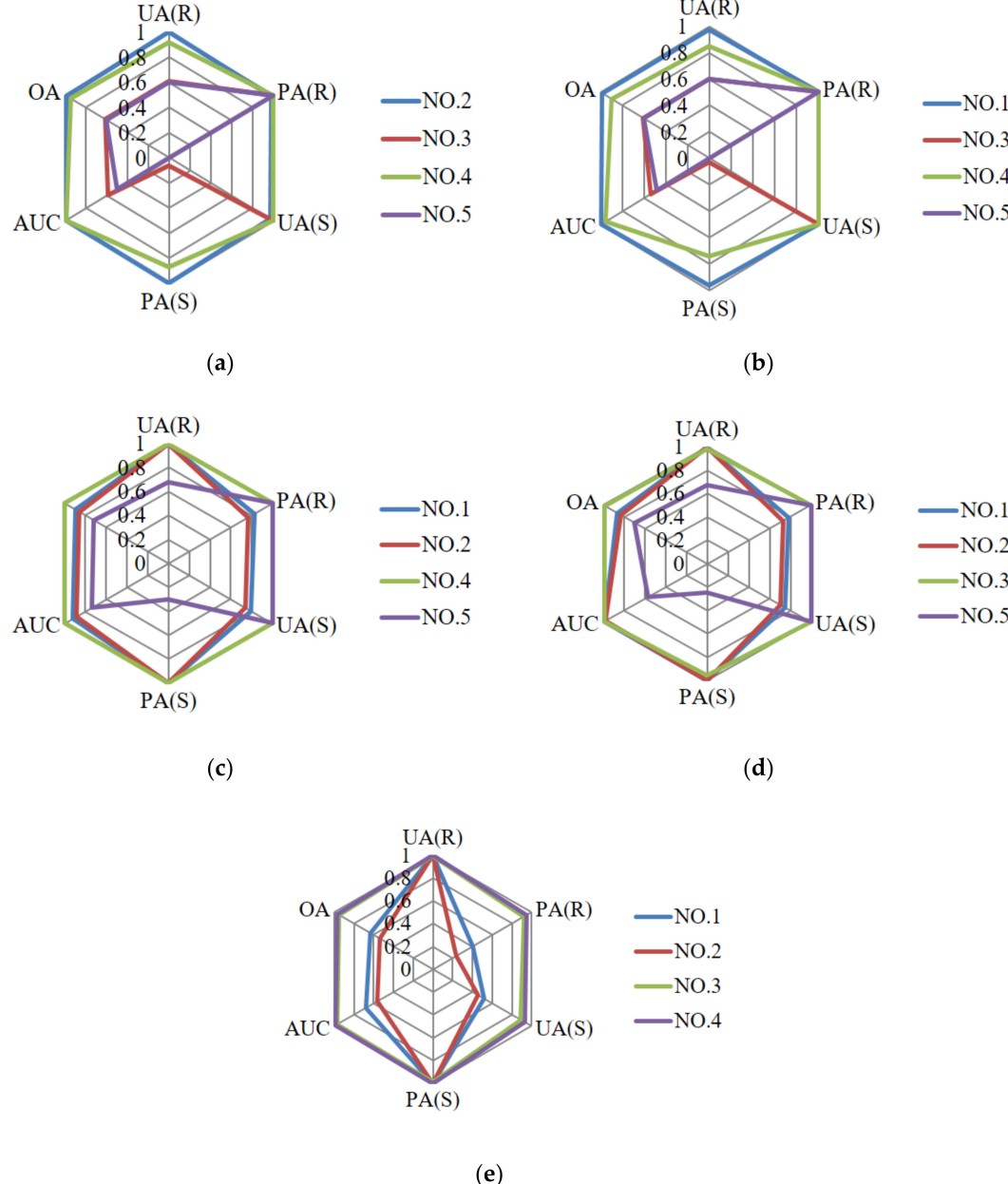

**Figure 6.** Accuracies of the random forest (RF)-based models using the top five polygon samples to predict each other, where OA, AUC, PA, UA, S, and R represent overall accuracy, the area under the ROC curve, producer's accuracy, user's accuracy, landslide source, and runout, respectively. (**a**) No. 1, (**b**) No. 2, (**c**) No. 3, (**d**) No. 4, and (**e**) No. 5 polygon samples for modeling.

To address this problem, cost-sensitive analysis procedures, similar to the procedure presented by Lai and Tsai [48], were employed to adjust the decision boundary during RF-based modeling. The results of the cost-sensitive analysis for different costs are illustrated in Figure 7. The developed model without the cost-sensitive analysis (i.e., the cost is equal to 1) had a high missing error for detecting landslide source regions, as displayed in Figure 7a–f, and runout areas, as displayed in Figure 7g,h, because of the assignment of higher FN and FP, respectively. The variations and improvements of accuracies could be observed after increasing the costs of FN and FP for the cases presented in Figures 7a–f and 7g,h, respectively. Based on Figures 6 and 7, Table 3 lists the representative models applying the RF algorithm and cost-sensitive analysis with different inventory polygon samples for further verification, as presented in Section 3.3.

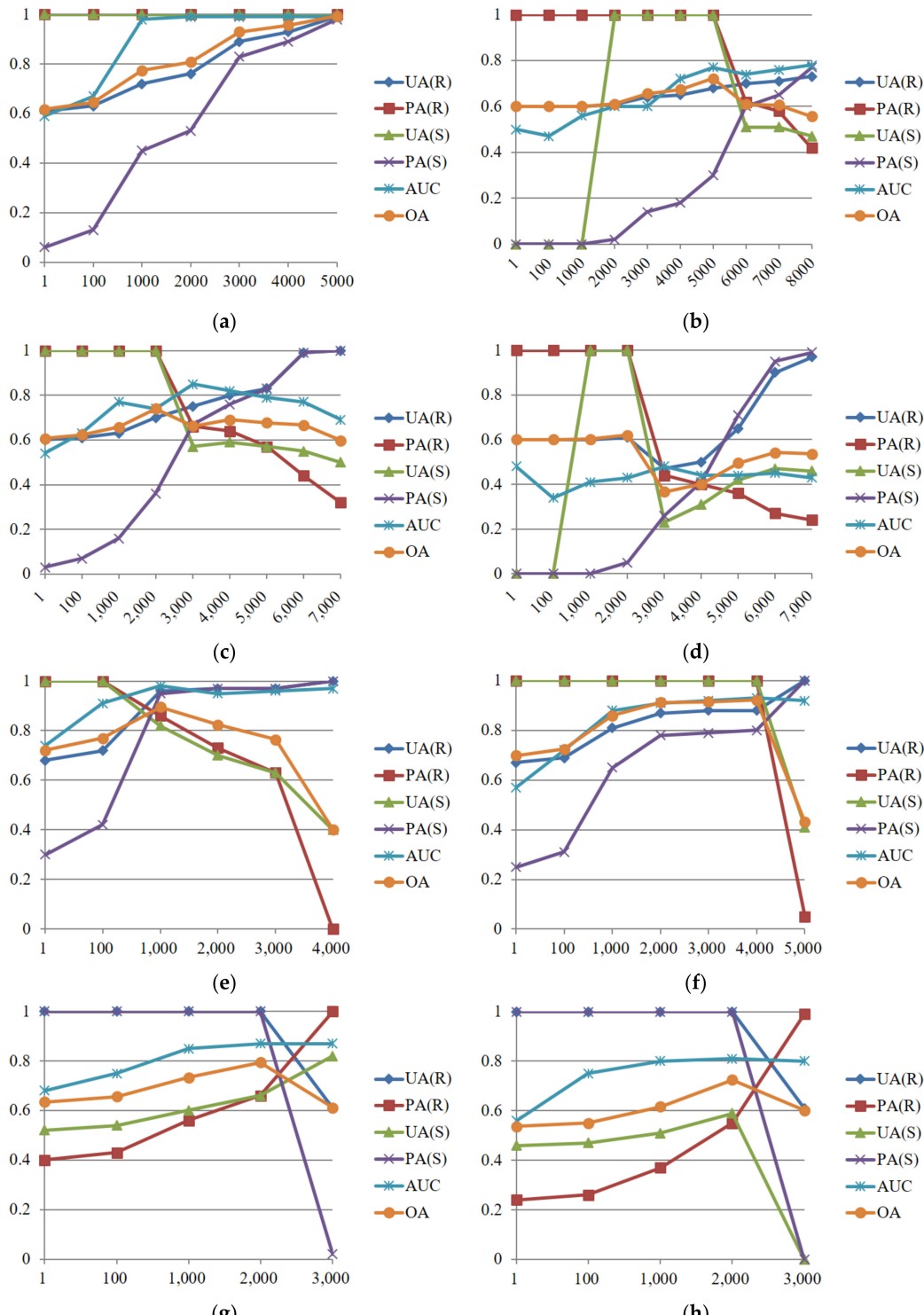

**Figure 7.** Accuracies (*Y*-axis) of the RF-based models with different costs (*X*-axis) and polygon samples for predictions, where OA, AUC, PA, UA, S, and R represent the overall accuracy, area under the ROC curve, producer's accuracy, user's accuracy, landslide source, and runout, respectively. (**a**) No. 1 model predicting No. 3 polygon samples, (**b**) No. 1 model predicting No. 5 polygon samples, (**c**) No. 2 model predicting No. 3 polygon samples, (**d**) No. 2 model predicting No. 5 polygon samples, (**e**) No. 3 model predicting No. 5 polygon samples, (**f**) No. 4 model predicting No. 5 polygon samples, (**g**) No. 5 model predicting No. 1 polygon samples, and (**h**) No. 5 model predicting No. 2 polygon samples.

**Table 3.** The representative models using the random forests (RF) algorithm with the costs for verifications based on pervious comparisons. For example, RF_3000 indicates that the RF algorithm with a cost of 3000 was used.

| Model | No. 1 | No. 2 | No. 3 | No. 4 | No. 5 |
|---|---|---|---|---|---|
| Algorithm_Cost | RF<br>RF_3000<br>RF_4000<br>RF_5000<br>RF_6000<br>RF_7000 | RF<br>RF_3000<br>RF_4000<br>RF_5000 | RF<br>RF_1000 | RF<br>RF_2000<br>RF_3000<br>RF_4000 | RF<br>RF_2000 |

### 3.3. Model Performances and Comparisons

Based on settings presented in Table 3, the constructed models were used to predict the samples extracted from other inventory polygons with an area of ≥100,000 or 50,000–100,000 m², namely the larger and smaller inventory polygons, presented in Sections 3.3.1 and 3.3.2, respectively. Furthermore, the performances between the developed models and the approaches of SVM and LR were compared.

### 3.3.1. The Case of Larger Inventory Polygons

The prediction results of No. 1 models obtained using the RF and cost-sensitive analysis are presented in Figure 8a. The examination of this figure reveals that a cost of 5000 provided more favorable results, with the accuracy reaching over 80% in most indices. However, the evaluations of SVM-based approaches in consideration of linear, the first- to third-order polynomial, and radial basis functions are displayed in Figure 8b. Overall, the second-order polynomial function outperformed others. Compared with previous LR results, Figure 8c indicates that the developed model in Section 3.2 obtained the highest accuracies, although the UA of the landslide source was slightly lower.

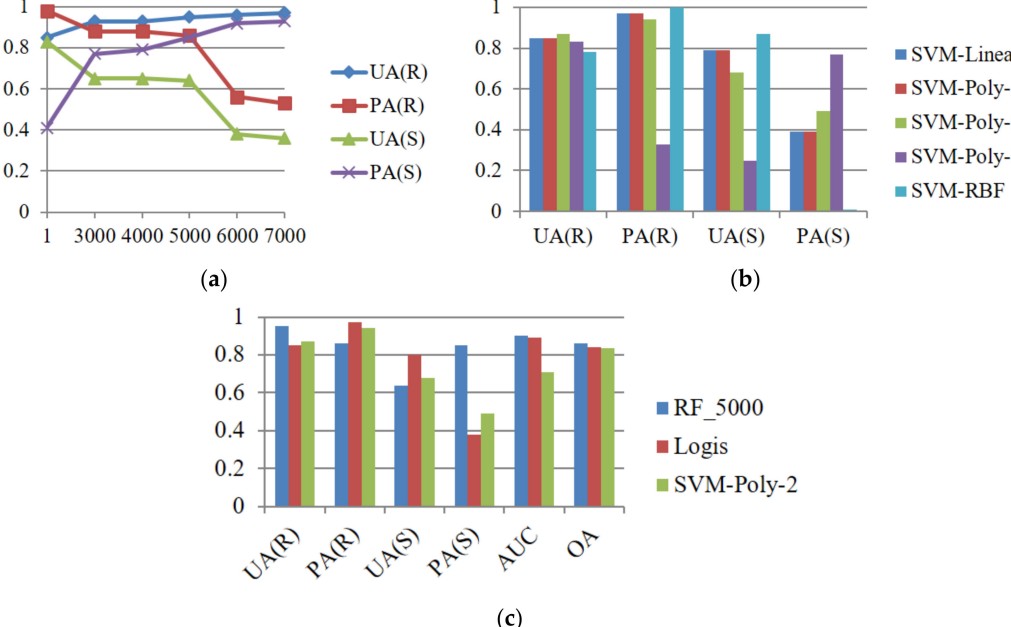

**Figure 8.** Performances (*Y*-axis) of No. 1 model, where OA, AUC, PA, UA, S, and R represent the overall accuracy, area under the ROC curve, producer's accuracy, user's accuracy, landslide source, and runout, respectively. (**a**) Using the algorithm with the costs; (**b**) using the support vector machine (SVM) with linear, the first- to third-order polynomial, and radial basis (RBF) functions; (**c**) comparing the optimal results of (**a**) and (**b**) by using logistic (Logis) regression.

Similar procedures to those used to verify the No. 2 model were applied to assess the algorithms of RF with the cost-sensitive analysis, SVM, and LR. Figure 9a further compares the RF with costs, as displayed in the third column in Table 3. Based on this figure, the RF algorithm appeared to outperform the other models without increasing costs. As illustrated in Figure 9b, the accuracies of all the functions of SVM modeling were similar, with higher missing errors for detecting landslide sources. The RF and the second-order polynomial function of SVM were compared with LR, as illustrated in Figure 9c, which revealed that the AUC of the LR was the highest. The accuracies of the three other approaches were similar.

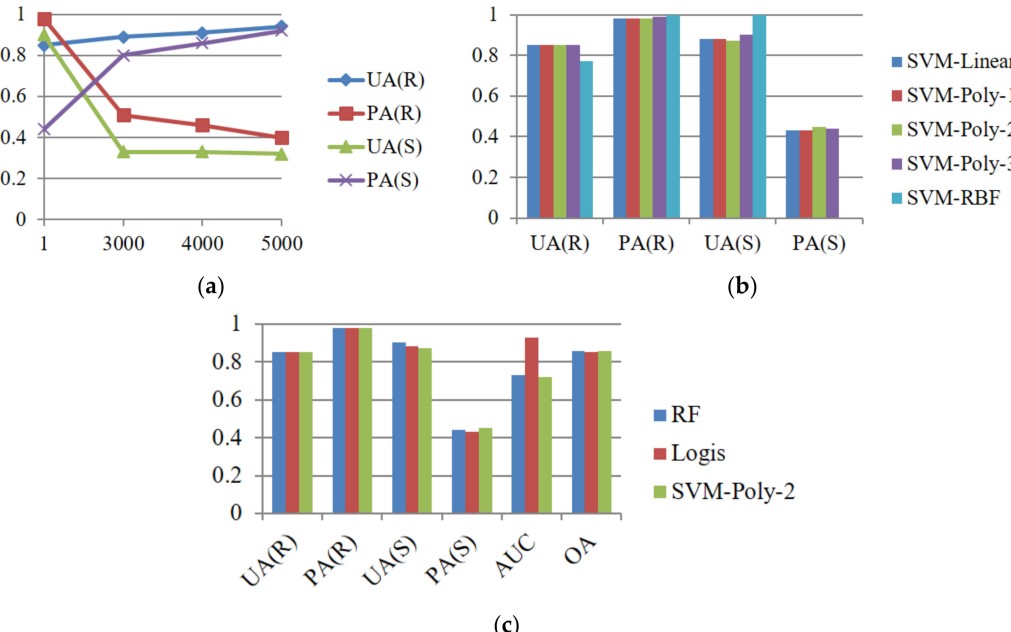

**Figure 9.** Performance (*Y*-axis) of the No. 2 model, where OA, AUC, PA, UA, S, and R represent the overall accuracy, area under the ROC curve, producer's accuracy, user's accuracy, landslide source, and runout, respectively. (**a**) Using the RF algorithm with the costs; (**b**) using SVM with linear, the first- to third-order polynomial, and radial basis (RBF) functions; (**c**) comparing the optimal results of (**a**) and (**b**) by using logistic (Logis) regression.

For the No. 3 model, Table 4 presents the comparison of the RF algorithm and cost-sensitive analysis. The results indicate that the original RF algorithms provided higher accuracy, whereas the RF method with a cost of 1000 had lower PA and UA for the runout and landslide source classes, respectively. In the SVM case, Figure 10a reveals that the RBF obtained the least accurate results, especially PA and UA for the runout and landslide source signatures. The performances of linear and the first-order polynomial functions were similar. A further comparison of these models, illustrated in Figure 10b, revealed that the results produced by the RF method were more favorable in most cases, especially UA for the landslide source class.

**Table 4.** The performances of the No. 3 model by using the RF method with the cost of 1000, where OA, AUC, PA, UA, S, and R represent overall accuracy, the area under ROC curve, producer's accuracy, user's accuracy, landslide source, and runout, respectively.

| Method | UA(R) | PA(R) | UA(S) | PA(S) | AUC | OA |
|--------|-------|-------|-------|-------|-----|-----|
| RF | 0.96 | 0.86 | 0.66 | 0.87 | 0.89 | 86.42% |
| RF_1000 | 1 | 0.56 | 0.41 | 0.99 | 0.92 | 66.39% |

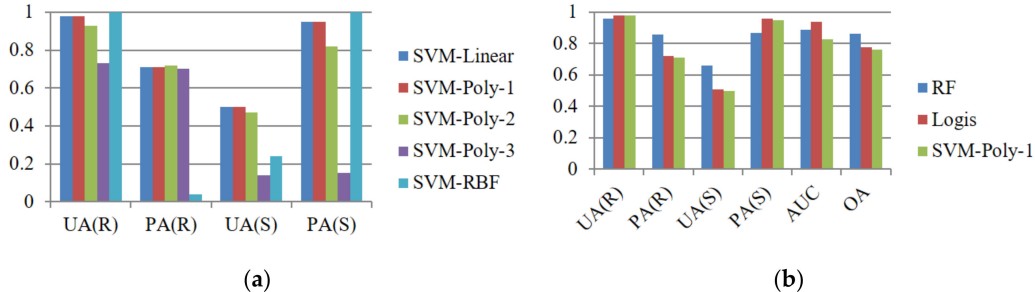

**Figure 10.** Performances (*Y*-axis) of the No. 3 model, where OA, AUC, PA, UA, S, and R represent the overall accuracy, area under the ROC curve, producer's accuracy, user's accuracy, landslide source, and runout, respectively. (**a**) Using SVM with linear, the first- to third-order polynomial, and radial basis (RBF) functions; (**b**) comparing the optimal results of Table 4 and (**a**) by using logistic (Logis) regression.

The No. 4 models were also examined as illustrated in Figure 11. The RF method without costs also provided the most accurate results, as presented in Figure 11a. Similarly, the performances of the linear and first-order polynomial functions outperformed the RBF method in SVM-based computation, as illustrated in Figure 11b. The comparison based on the aforementioned methods is illustrated in Figure 11c. The findings indicated that the AUC result of SVM was slightly higher than the other indices of the RF, SVM, and LR, which were similar.

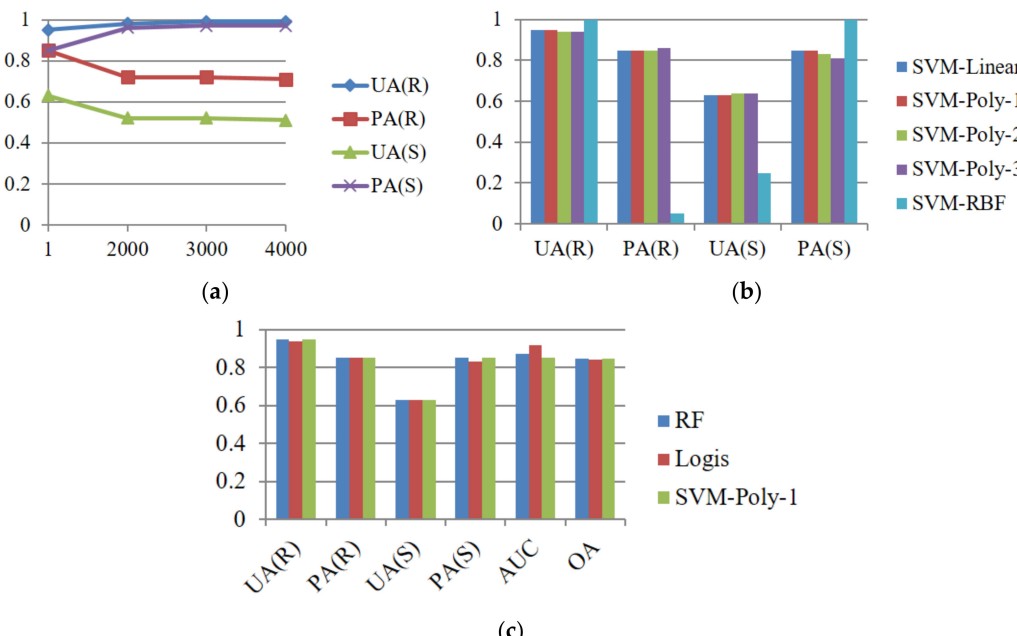

**Figure 11.** Performance (*Y*-axis) of the No. 4 model, where OA, AUC, PA, UA, S, and R represent overall accuracy, area under the ROC curve, producer's accuracy, user's accuracy, landslide source, and runout, respectively. (**a**) Using the RF algorithm with the costs; (**b**) using SVM with linear, the first- to third-order polynomial, and radial basis (RBF) functions; (**c**) comparing the optimal results of (**a**) and (**b**) with logistic (Logis) regression.

In the No. 5 model, the results produced by the RF algorithm with a cost of 2000 displayed high accuracy, as illustrated in Table 5. For SVM-based modeling, Figure 12a indicated that the RBF method had unbalanced predictions with a significant missing error for the landslide source and the highest PA for the runout. However, the results of the linear and polynomial functions were similar. The second-order polynomial SVM and LRs, illustrated in Figure 12b, displayed accuracies similar to the RF_2000.

**Table 5.** Performance of the No. 5 model using the RF method with a cost of 2000, where OA, AUC, PA, UA, S, and R represent the overall accuracy, area under the ROC curve, producer's accuracy, user's accuracy, landslide source, and runout, respectively.

| Method | UA(R) | PA(R) | UA(S) | PA(S) | AUC | OA |
|--------|-------|-------|-------|-------|-----|-----|
| RF | 1 | 0.49 | 0.38 | 1 | 0.76 | 61.44% |
| RF_2000 | 1 | 0.76 | 0.56 | 1 | 0.91 | 81.46% |

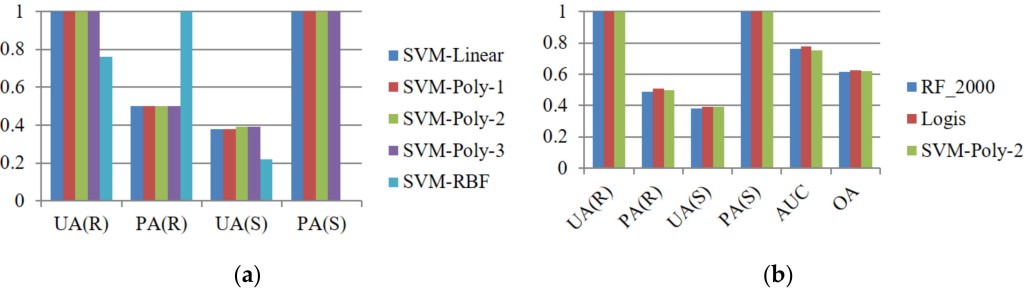

(a)　　　　　　　　　　　　　　　　　　　　(b)

**Figure 12.** Performances (Y- axis) of the No. 5 model, where OA, AUC, PA, UA, S, and R represent the overall accuracy, area under ROC curve, producer's accuracy, user's accuracy, landslide source, and runout-out, respectively. (**a**) Using SVM with linear, the first- to third-order polynomial, and radial basis (RBF) functions, (**b**) comparing the optimal best results of Table 5 and (**a**) with logistic (Logis) regression.

Significant improvements were observed in the No. 1 and No. 3 models. Moreover, the RF algorithm and cost-sensitive analysis provided favorable results in most cases. To identify the optimal model in which the samples extracted from different size areas of the inventory polygons, Figure 13a compares the RF method for the No. 2, 3, and 4 models, with costs of 5000 and 2000 for No. 1 and 5 models, respectively. Figure 13 indicates that the accuracies produced by the No. 1, 3, and 4 models were similar, whereas the No. 2 and 5 models displayed less favorable results for the detection of a landslide source class. Figure 13b indicated that the AUC of the No. 1 and 3 models were the most favorable. These evaluations indicated that the No. 1 and 3 models provided more reliable prediction results.

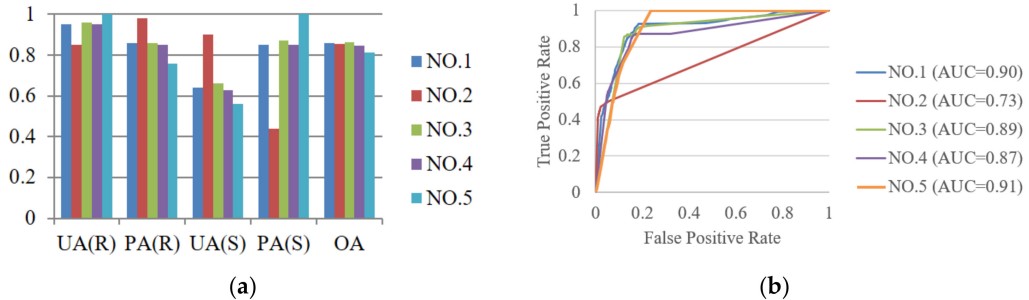

(a)　　　　　　　　　　　　　　　　　　　　(b)

**Figure 13.** Comparison of the representative results using the RF method for the No. 2, 3, and 4 models, with costs of 5000 and 2000 for No. 1 and 5 models, respectively. (**a**) User's accuracy (UA), producer's accuracy (PA), and overall accuracy (OA); (**b**) area under the ROC curve (AUC) for landslide source (S) and runout (R) detections.

### 3.3.2. The Case of Smaller Inventory Polygons

The performances of the No. 1 model based on the RF method with a cost of 5000 and the No. 3 model with the RF algorithm were assessed, as presented in Table 6, to predict the samples extracted from the polygon areas equal to or larger than 50,000 and smaller than 100,000 m$^2$. The accuracies were lower than those displayed in Figure 13a, especially for the runout UA and PA of

the No. 1 model. Furthermore, the No. 3 model with the RF method outperformed the No. 1 model, indicating accurate results. The topographic roughness index (TRI) is commonly used for assessing landslide susceptibilities [61]. TRI was further used to improve this case as also listed in Table 6. However, the significant topographic attributes [62–64] and various indices [65] for separating landslide source and runout signatures need to be further explored. These results reveal the uncertainty between the training data and prediction targets. More precisely, the area size of the training inventory polygon affected the capability and applicability of the constructed models. In practice, manually interpreting the larger targets of landslide source and runout from remotely sensed images to produce the training data was easier than selecting smaller objects, which was a clear limitation of using the supervised strategy with four topographic variables in this study. Addressing this problem requires further consideration regarding the tradeoff between the cost of producing training data and the applicability of the developed models. The developed models were suggested for different area sizes of the inventory polygons.

**Table 6.** Performances of the No. 1 model using the RF method with a cost of 5000 and No. 3 model with the RF algorithm, where OA, AUC, PA, UA, S, and R represent the overall accuracy, area under the ROC curve, producer's accuracy, user's accuracy, landslide source, and runout, respectively.

| Model | Method | UA(R) | PA(R) | UA(S) | PA(S) | AUC | OA |
|-------|--------|-------|-------|-------|-------|-----|-----|
| No. 1 | RF_5000 | 0 | 0 | 0.64 | 1 | 0.67 | 63.87% |
| No. 3 | RF | 0.59 | 0.93 | 0.94 | 0.63 | 0.85 | 73.94% |
| No. 3* | RF | 0.63 | 0.91 | 0.93 | 0.70 | 0.85 | 77.54% |

\* With the topographic roughness index.

### 3.4. Visualization of Landslide Detection

To visualize the separated results of the case in Section 3.3.1, the predicted patterns of the No. 3 models for distinguishing landslide source and runout targets were generated, as illustrated in Figure 14. The black and red and blue colors in this figure represent the ground truth and prediction results, respectively. As illustrated in Figure 14a, the developed models displayed suitable detection of the landslide sources (high PA(S)) with some false alarm errors (low UA(S)). Figure 14b also illustrated that the model provided reliable results for detecting runout areas with higher UA(R) and PA(R). These patterns accorded with the results of the No. 3 model in Figure 10b.

The detected distributions with the topographic situation in a three-dimensional (3D) space were further examined. Figure 15 displays the patterns of the No. 3 model for predicting the No. 1 and No. 2 landslide source and runout inventory polygons. In this figure, red and yellow represent the correct and incorrect predictions, respectively. Figure 15a,b displays accurate detections for the landslide sources. The results presented in Figure 15c indicate that the lower part of a runout area can be detected, whereas the upper region may be miscategorized as a landslide source. This finding reveals that elevation is a critical factor for the separation of landslide source and runout areas. This finding also accords with the result of the factor analysis presented in Section 3.1. Future studies can determine the elevation-based threshold to improve the capability of the developed models.

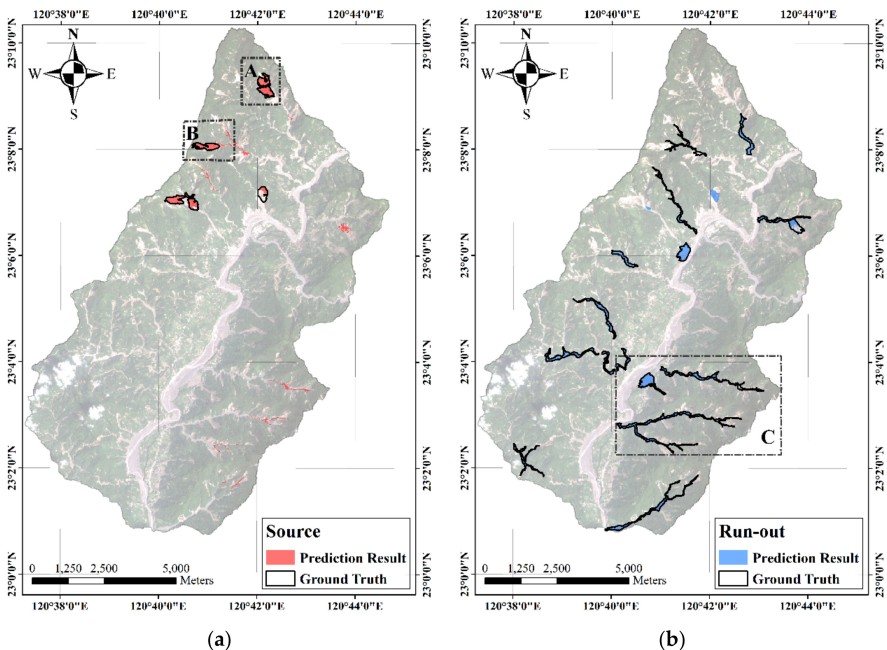

(a)  (b)

**Figure 14.** Prediction results of the No. 3 model using the RF algorithm to separate landslide source and runout signatures with the areas of inventory polygons of ≥100,000 m². Cases of (**a**) landslide source and (**b**) runout polygons.

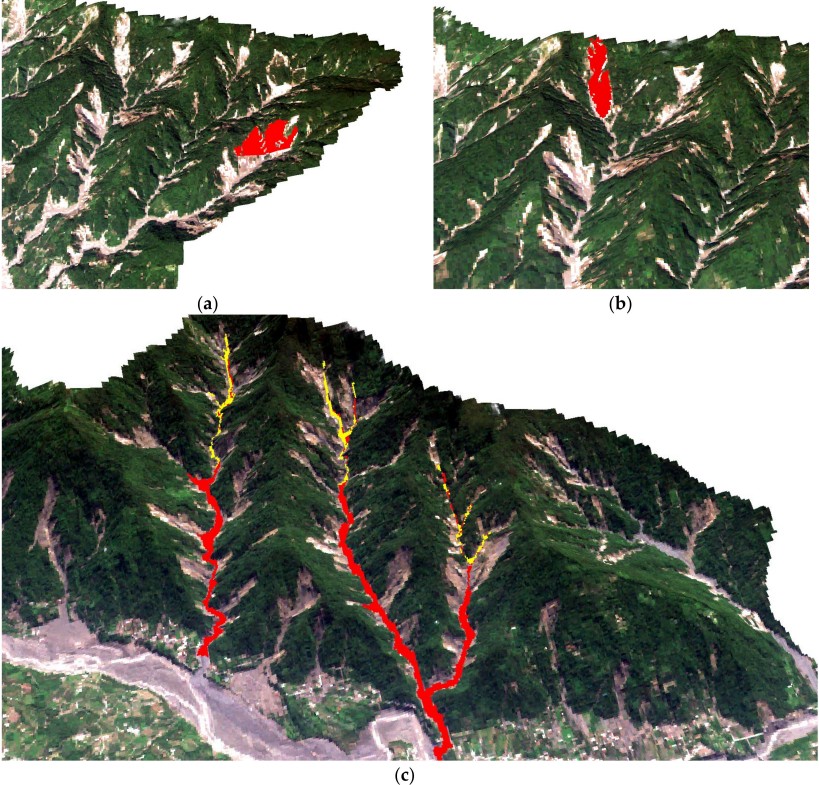

**Figure 15.** Three-dimensional (3D) view of the No. 3 model for predictions of separating landslide source and runout areas. The landslide source areas of the (**a**) No. 1 and (**b**) No. 2 polygons in the landslide inventory (located region A and B in Figure 14), and (**c**) the runout regions of the No. 1 and No. 2 inventory polygons (located region C in Figure 14). Red and yellow represent correct and incorrect results, respectively.

## 4. Discussion

The major finding in this study was that detecting landslides is not fully equal to inventory generation. The detected targets require further examination. Lai et al. [34] suggested that the runout area and topological relationship could be considered independent classes in landslide records when producing a landslide inventory. Therefore, this study focused on the separation of landslide source and runout classes caused by different mechanisms. The feasibility of distinguishing between landslide source and runout signatures to improve landslide detections was assessed. The limited training samples extracted from the top five polygons in the landslide inventory map were used to construct supervised data mining models in order to classify other instances. The top five polygon samples were used for training because manually recognizing the larger targets from a regional scale-based remote sensing image is relatively easier than interpreting smaller objects. Based on this scenario, the developed models with RFs and cost analysis and mapping results demonstrated the stability of the proposed procedure. The hierarchical analysis was designed in this study. Section 3.2 explored the feasible models as listed in Table 3. Section 3.3 demonstrated the performances of developed models verified by the samples of larger and smaller inventory polygons, comparing them with the SVM and LR methods as shown in Figures 8–13 and Tables 4–6. The quantitative evaluations of the optimal results as shown in Figure 13 were determined to be higher than 80% in most cases. The results further demonstrated that the developed models outperform the SVM and LR algorithms in most accuracies.

The primary limitation of this study was that the developed models were more favorable for larger area sizes but may be difficult to apply to smaller areas. The capability of the models based on the supervised strategy was related to the characteristics of the training datasets. The tradeoff between the cost of producing the training data and the area sizes of targets for predictions should be considered. Perhaps a multi-scale model trained by different area sizes of the inventory polygons is a possible solution. This method could also be improved by including more topographic attributes, as described by Sallem et al. [61].

Producing a high-quality landslide inventory is time consuming and expensive. Landslide inventories are crucial for landslide analyses, such as for performing tasks in the framework of landslide risk assessment and management [14]. The potential for reducing the degree of manual interpretation is demonstrated in mapping results. Based on the spatial patterns of predictions, the misclassification for detecting runout areas appeared in the top region. Two possible strategies may solve this problem. First, a threshold based on the elevation attribute could be designed. Second, studies could investigate only extracted landslide sources from the affected areas, and the remainder are assigned the runout label. These runout patterns may be adequate to further assess the related parameters, such as runout distance and damage corridor width [14].

## 5. Conclusions

A novel data analysis based on the scenario of separating landslide source and runout areas with the topographic attributes was presented in this paper to improve landslide detections. To explore the feasibility of the proposed approach, four hierarchical experiments were designed. First, elevation was a critical variable according to the factor analysis. Second, the candidate models were constructed using the RF and cost-sensitive analysis and the limited training samples extracted from the top five largest areas of the inventory polygons. The polygon-based cross-validation was also used for the verification. Third, the candidate models verified using different polygon-based samples indicated that the No. 1 model using the RF method with a cost of 5000 and the No. 3 model based on the RF were the most favorable for predicting the larger polygon (area $\geq$ 100,000 m$^2$) samples. For the smaller polygon (area of 50,000–100,000 m$^2$) samples, the No. 3 model provided acceptable results. The accuracies of developed models were above 0.8 in most cases, outperforming the performances of the SVM and LR. Finally, mapping the predicted distributions revealed that the detection of landslide sources provided accurate results with lower missing rates. The extraction for the runout areas revealed excellent user's and producer's accuracies.

The gap in recognizing landslides has existed in terms of remote sensing and geotechnical engineering. More precisely, the landslide detections extracted from remotely sensed images without additional interpretations to eliminate the runout areas should be termed the landslide affected area and should not be directly treated as a landslide inventory or database for further analyses. This study provided a systematic procedure following a supervised data mining–based approach to separate landslide source and runout signatures. The tradeoff between the size areas of the training data and the predicted range of the developed models was emphasized in the results. More research is warranted regarding the effect of the elevation-based threshold and the binary classification based on the detections of landslide sources only to produce fully automatic landslide inventory productions.

**Funding:** This study was supported, in part, by the Ministry of Science and Technology of Taiwan under project No. 109-2121-M-035-001 and 108-2119-M-035-003.

**Acknowledgments:** The author would like to thank S.-H. Chiang in Center for Space and Remote Sensing Research, National Central University, Taiwan, for providing the landslide inventory and the digital elevation model. The author also thanks J.-Y. Huang in Department of Civil Engineering, Feng Chia University, Taiwan, for refining the figures in the article.

**Conflicts of Interest:** The author declares no conflicts of interest.

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
