# Peer review of "Separating Landslide Source and Runout Signatures with Topographic Attributes and Data Mining to Increase the Quality of Landslide Inventory"

_applsci, doi:10.3390/app10196652_

Round 1

Reviewer 1 Report

Designed and performed variant analyses is always a difficult task due to the difficulty of presenting the results of the studies.

The Author graphically presented the results and described them but it is difficult to know what variant it is about (too many abbreviations of models and training grounds were used).

In the reviewer's view, the results for each option should be put together and the different options should be compared in chapter 4. Discussion.

Specific notes:

line 99: 117m2 ????

full text: consider using a comma in numeric values-compare line 99 and figure 1 , see table 1, see Figur 2-not acceptable

Figure 7 it is not clear in which units the values on the axes are given and what are the values 1, 100, 1,0

Table 3 is not clear

Reviewer 2 Report

Review of the manuscript submitted to Applied Science titled: ” Separating Landslide Source and Runout Signatures with Topographic Attributes and Data Mining to Improve Landslide Detection”.

The paper deals with separating landslide source from runout signature. The paper appears interesting, but the presented method is not a new method.. Generally paper is well written, experiment is designed and performed correctly. However my main criticism is represented by small exploration of DEM data. Nowadays, many derivatives or topographical factors are used for landslide detection e.g. roughness, openness etc. In my opinion, author should et least include roughness index to differentiate landslide source from runout signature.

Additionally, the explanation of “runout signature” should be presented in text or as figure. I can deduce what author means with runout areas but this terminology is not widely used. Widely used terminology include landslide main scarp, body, flanks, toe etc. Therefore, in my opinion author should refer to this terminology in order to make his/her paper clearly understood.

Moreover, other criticism goes to the aim of the study represented also in title “…to Improve Landslide Detection”. Author did not performed landslide detection. Authors used existing landslide database and some of landslides form this database and for this landslide author tried to differentiate source from runout. Detecting landslides take please when we try to identify landslide in the areas where we don’t have landslide inventory. But masking non landslide areas and making differentiation for landslide form landslide inventory is not a detection. I would rather call is as : Separating Landslide Source and Runout Signatures with Topographic Attributes and Data Mining towards detailed landslide inventory/to increase the quality of landslide inventory.

In the present form, the paper needs of a major revision before to be considered for publication in Applied Science

Minor observations are reported in the attached pdf file.

Reviewer

Round 2

Reviewer 2 Report

Please find review comments in attached file.
